# Pyridylpiperazine-based allosteric inhibitors of RND-type multidrug efflux pumps

Coline Plé[1,6], Heng-Keat Tam [2,5,6], Anais Vieira Da Cruz[3], Nina Compagne[3], Juan-Carlos Jiménez-Castellanos[1], Reinke T. Müller[2], Elizabeth Pradel [1], Wuen Ee Foong [2], Giuliano Malloci [4], Alexia Ballée[3], Moritz A. Kirchner[2], Parisa Moshfegh[3], Adrien Herledan[3], Andrea Herrmann[2], Benoit Deprez[3], Nicolas Willand[3], Attilio Vittorio Vargiu [4], Klaas M. Pos [2,7 ✉], Marion Flipo [3,7 ✉] & Ruben C. Hartkoorn [1,7 ✉]

Efflux transporters of the RND family confer resistance to multiple antibiotics in Gram-negative bacteria. Here, we identify and chemically optimize pyridylpiperazine-based compounds that potentiate antibiotic activity in *E. coli* through inhibition of its primary RND transporter, AcrAB-TolC. Characterisation of resistant *E. coli* mutants and structural biology analyses indicate that the compounds bind to a unique site on the transmembrane domain of the AcrB L protomer, lined by key catalytic residues involved in proton relay. Molecular dynamics simulations suggest that the inhibitors access this binding pocket from the cytoplasm via a channel exclusively present in the AcrB L protomer. Thus, our work unveils a class of allosteric efflux-pump inhibitors that likely act by preventing the functional catalytic cycle of the RND pump.

[1] Univ. Lille, CNRS, Inserm, CHU Lille, Institut Pasteur Lille, U1019—UMR 9017—CIIL—Center for Infection and Immunity of Lille, F-59000 Lille, France. [2] Institute of Biochemistry, Goethe-University Frankfurt, Max-von-Laue-Str. 9, D-60438 Frankfurt am Main, Germany. [3] Univ. Lille, Inserm, Institut Pasteur de Lille, U1177—Drugs and Molecules for Living Systems, F-59000 Lille, France. [4] Department of Physics, University of Cagliari, 09042 Monserrato (Cagliari), Italy. [5] Present address: Hengyang Medical School, University of South China, 421002 Hengyang, Hunan Province, China. [6] These authors contributed equally: Coline Plé, Heng-Keat Tam. [7] These authors jointly supervised this work: Klaas M. Pos, Marion Flipo, Ruben C. Hartkoorn. ✉email: pos@em.uni-frankfurt.de; marion.flipo@univ-lille.fr; ruben.hartkoorn@inserm.fr

Infections by multidrug-resistant (MDR) Gram-negative bacteria are a major threat to global healthcare and are currently classified as a critical priority for R&D by the World Health Organization. Among many antibiotic-specific resistance mechanisms, active antibiotic extrusion by multidrug efflux pumps is a major contributor to both intrinsic (basal) and acquired multiple drug resistance phenotypes in Gram-negative bacteria. In particular, efflux pumps belonging to the Resistance Nodulation cell Division (RND) superfamily extrude a plethora of chemically diverse antibiotic molecules and represent a major barrier to antibiotic efficacy and drug development[1–3].

RND-type efflux pumps span the entire Gram-negative envelope[4–7] and are composed of a proton motive force-driven inner membrane transporter (e.g. AcrB), a periplasmic adaptor protein (e.g. AcrA), and an outer membrane channel (e.g. TolC)[8]. Homotrimeric AcrB is comprised of three protomers that cycle interdependently through the loose (L), tight (T), and open (O) conformational states during H$^+$/drug antiport catalysis[9–12]. Efflux pump substrates enter AcrB via dedicated entrance channels toward the proximal access pocket (AP) in the L protomer or toward the more distal deep binding pocket (DBP) exclusively present in the T protomer. Substrate binding to this DBP leads to structural rearrangements that propagate into the transmembrane (TM) region to allow for protonation of titratable side chains (D407 and D408) located deeply inside the TM domain[13]. The electrostatic changes upon protonation lead to the transition from the T state to the O state, resulting in the collapse of the entrance tunnels and the DBP. Concurrently, the substrate leaves the DBP via a newly formed exit tunnel in the AcrB O protomer and is transported through the AcrA-TolC channel across the outer membrane. The AcrB O protomer subsequently resets to the L state through a proton motive force-dependent H$^+$-release from the protonated Asp pair into the cytoplasm[8,9,12,14,15].

Considering the critical role of RND pumps in innate and acquired antibiotic resistance in Gram-negative bacteria, there have been considerable efforts to discover and develop efflux pump inhibitors (EPI). Characterized EPIs include Phe-Arg β-naphthylamide (PAβN) and 1-(1-naphthylmethyl)-piperazine (NMP), which inhibit RND-efflux by binding and blocking the DBP or competitively preventing antibiotic binding to this site. More specific allosteric EPIs such as pyridopyrimidines (such as D13-9001), and pyranopyridines (the MBX series) bind to a hydrophobic pocket (aka "hydrophobic trap") proximal to the DBP of E. coli AcrB[16–23], halting the catalytic efflux cycle by blocking the T to O transition, while likely also competing with drug binding in the DBP[22,24].

In this work, we sought to discover a novel chemical scaffold able to boost antibiotic activity in E. coli by inhibiting the AcrAB-TolC efflux pump. In line with this goal, a pyridylpiperazine-based molecule was discovered by phenotypic screening and was validated to act specifically on AcrB. Medicinal chemistry uncovered more potent inhibitors and structural biology showed the location of the pyridylpiperazine binding pocket in the AcrB TM domain. Our findings bring to light a novel class of EPIs that act in a previously unexploited pocket, which likely allosterically prevents protomer cycling and the drug efflux process. Together, the presented work offers an alternative prospect in the fight against antibiotic resistance of Gram-negative bacteria.

## Results

**Identification of new EPIs**. To identify EPIs, a phenotypic assay was established to screen a chemical library of 1280 fragments (Table S1). The molecules selected in this chemical library (originating from both commercially available molecules and unique in-house synthesized compounds) were enriched with C-sp³,

spiro carbon atoms and halogen atoms[25,26]. Additionally, they fulfilled the commonly accepted 'rule of three' (MW < 300 g/mol, cLogP < 3, number of H-bond donors ≤ 3, number of H-bond acceptors ≤ 3)[27]. The EPI screening assay searched for molecules that synergized the activity of a sub-MIC (Minimal Inhibitory Concentration) dose of the antibiotic pyridomycin[28], a particularly good substrate of the AcrAB-TolC efflux pump (MIC for E. coli BW25113 is 12.5–25 µg/mL whereas for E. coli BW25113 ΔacrB or ΔtolC mutants it is 0.78 µg/mL). Screening of the fragment library at 300 µM identified a one-hit molecule, BDM73185 (1), which prevented E. coli BW25113 growth in the presence of sub-MIC pyridomycin (5 µg/mL) but exhibited no antibiotic activity by itself. BDM73185 (1) is a small soluble molecule (MW: 265 g/mol, PBS solubility > 900 µM) with a pyridine core, two electron-withdrawing substituents (Cl and CF₃) in positions 3 and 5 and a basic piperazine moiety in position 2 (Table 1). In validation experiments, BDM73185 (1) (300 µM) boosted the antibiotic activity of a panel of AcrAB-TolC substrates such as chloramphenicol (8-fold), pyridomycin (4-fold), tetracycline (4-fold), erythromycin and ciprofloxacin (2-fold), but did not alter the activity of non-AcrB substrates streptomycin and kanamycin (Table S2). In addition, BDM73185 (1) did not mediate antibiotic boosting in E. coli ΔacrB, ΔacrA, or ΔtolC mutants (Table S3).

**Spontaneous AcrB point mutations confer BDM73185 resistance**. To validate AcrAB-TolC as the BDM73185 (1) target, spontaneous E. coli mutants resistant to BDM73185 (1)-mediated antibiotic boosting were selected, without changing the basal antibiotic susceptibility in the absence of the inhibitor. Selection of E. coli BW25113 on solid media containing 600 µM BDM73185 (1) and different sub-inhibitory concentrations of erythromycin (1.25, 2.5, or 5 µg/mL) resulted in the selection of 2, 1, and 1 resistant colony, respectively (frequency of resistance ~1.5 × 10⁹). The four selected resistant isolates showed no change in basal susceptibility to either erythromycin or pyridomycin (confirming no change in the basal activity of the AcrAB-TolC efflux pump), and this antibiotic susceptibility was no longer boosted by BDM73185 (1) (Table S4). Targeted Sanger sequencing found that all four resistant isolates carried a nonsynonymous mutation of acrB coding for proximal residues in the 5th TM helix (TM5) of AcrB (3 mutants with t1348c [S450P], and one with g1336c [A446P]). Whole-genome sequencing and variant analysis confirmed that no other mutations were selected in these BDM73185 (1) resistant mutants, implying that the resistance phenotype selected was linked solely to an acrB point mutation. These mutations in TM5 are spatially distant from the periplasmic AcrB regions targeted by known EPIs such as PAβN[29], MBX series[19] and NMP[30]. Concordantly, the canonical AcrB inhibitors PAβN (15 µg/mL) and NMP (250 µg/mL) were found to still boost pyridomycin antibiotic activity in the BDM73185 (1) resistant strains (Table S5). As an additional validation, a reverse genetics approach by recombineering was used to specifically reintroduce wildtype acrB, acrB t1348c [S450P], or acrB g1336c [A446P] into its natural chromosomal locus in E. coli ΔacrB, and these engineered strains showed the same BDM73185 (1) resistance phenotype as the selected strains (Table S6). Together, these data imply that the identified acrB mutations are solely responsible for the BDM73185 (1) resistance phenotype.

**Medicinal chemistry**. While BDM73185 (1) was the cornerstone in discovering this class of EPIs, it lacked potency and was unable to revert pyridomycin efficacy to that of E. coli ΔacrB and E. coli ΔtolC. Therefore, structure-activity relationships (SAR) around

**Table 1 Biological activities of compounds 1–12.**

| Compound | A | R1 | R2 | R3 | EPI antibiotic activity MIC$_{90}$ (µM)[a] | EPI antibiotic boosting activity EC$_{90}$ (µM)[b] |
|---|---|---|---|---|---|---|
| **1** (BDM73185) | **NH** | Cl | CF$_3$ | H | > 500 | 62 ± 21 |
| **2** | **O** | Cl | CF$_3$ | H | > 500 | > 500 |
| **3** | **CH$_2$** | Cl | CF$_3$ | H | > 500 | > 500 |
| **4** | **N-CH$_3$** | Cl | CF$_3$ | H | > 500 | 375 ± 125 |
| **5** | NH.HCl | Cl | **CH$_3$** | H | > 500 | 62 |
| **6** | NH.HCl | Cl | **OCH$_3$** | H | > 250 | 94 ± 32 |
| **7** | NH.HCl | Cl | **Br** | H | > 500 | 32 |
| **8** (BDM88832) | NH | Cl | **I** | H | > 500 | 12 ± 4 |
| **9** (BDM88855) | NH | **Cl** | | | > 250 | 3.4 ± 0.7 |
| **9'** (BDM88855.HCl) | NH.HCl | **Cl** | | | > 250 | 3.6 ± 0.5 |
| **10** | NH.HCl | **H** | | | > 250 | > 250 |
| **11** | NH.HCl | **Br** | | | > 250 | 1.5 ± 0.5 |
| **12** | NH.HCl | **I** | | | > 250 | 3 ± 1 |

[a]MIC$_{90}$ represents the minimal inhibitory concentration of tested compounds that prevent 90% of *E. coli* BW25113 growth as measured by resazurin reduction.
[b]EC$_{90}$ represents the effective concentration of tested compounds that prevents the growth of *E. coli* BW25113 in the presence of 8 µg/mL pyridomycin as measured by resazurin reduction (the MIC$_{90}$ of pyridomycin alone is 12.5–25 µg/mL). Data are the result of at least two biological replicates and are presented as mean values ± SEM. Source data are provided as a Source Data file.

the pyridylpiperazine hit were investigated to generate more potent analogues. This SAR study interrogated the role of the piperazine ring, the trifluoromethyl group and the chlorine atom of BDM73185 (**1**) (Table 1).

SAR analysis of the piperazine ring showed that the basic nitrogen was necessary for pyridomycin boosting activity in *E.*

*coli*, as its replacement by oxygen (**2**) or a carbon atom (**3**) resulted in inactive compounds (EC$_{90}$ > 500 µM). Addition of a methyl group on the basic nitrogen (**4**) led to a 6-fold loss in potency (EC$_{90}$ = 375 µM) suggesting a possible steric hindrance. Replacement of the hydrophobic trifluoromethyl group with a more polar methoxy group (**6**, EC$_{90}$ = 94 µM) caused a slight

decrease in activity. In contrast, replacement with a hydrophobic methyl group (**5**, $EC_{90} = 62\,\mu M$) did not impact compound potency, suggesting that hydrophobicity is an important criterion for activity. This was confirmed by the improved potency observed with the introduction of bromine (**7**, $EC_{90} = 32\,\mu M$) or an iodine atom (BDM88832 (**8**), $EC_{90} = 12\,\mu M$) instead of the trifluoromethyl group. It was therefore decided to replace the 5-iodopyridine with a quinoline ring (BDM88855 (**9**) and BDM88855.HCl (**9'**)), leading to a 15-fold more potent compound than the initial hit. Finally, removal of the chlorine atom (R1 group) of the quinoline ring significantly decreased activity (**10**, $EC_{90} > 250\,\mu M$), whereas its replacement by other halogen atoms was well tolerated (**11** and **12**, $EC_{90} = 1.5\,\mu M$ and $3\,\mu M$, respectively). None of the compounds showed antibacterial activity by themselves at the highest concentration tested. Overall, this SAR study led to the identification of fragment-sized pyridylpiperazine derivatives (compounds **9**, **11**, and **12**) 15 to 30 times more potent than the initial hit. Both BDM73185 (**1**) and BDM88855.HCl (**9'**) did not induce apoptosis ($CC_{50} > 100\,\mu M$) in cytotoxicity assays on BALB/3T3 cells following 24 and 48 h of exposure and were therefore considered non-cytotoxic in vitro.

**The spectrum of antibiotic boosting by BDM88855.** The activity of a large panel of antibiotics was determined on wild-type *E. coli* in the absence and presence of 100 μM BDM88855.HCl (**9'**). The antibiotic activity was also determined on *E. coli ΔacrAB*, which served as a proxy for full AcrB inhibition, helping clearly define the AcrB antibiotic substrates (Table 2). Data confirmed that all the AcrB substrates antibiotics were efficiently boosted by BDM88855.HCl (**9'**) in wild-type *E. coli*, to levels similar to that observed for *E. coli ΔacrAB*. Parallel experiments evaluating this antibiotic boosting by BDM88855.HCl (**9'**) in the pyridylpiperazine resistant *E. coli* isolates (with AcrB mutations S450P or A446P), showed these isolates to be cross-resistant to BDM88855.HCl (**9'**), confirming that these analogues of the initial hit retained activity on AcrB (Table S7).

**Checkerboard assay to evaluate antibiotic boosting.** To compare the efficacy of BDM88855.HCl (**9'**) and PAβN, their concentration-dependent boosting of pyridomycin and oxacillin activity was determined using a checkerboard assay. Data show a concentration-dependent modulation of antibiotic activity by BDM88855.HCl (**9'**), with boosting observed from 3.1–6.25 μM, reaching maximal inhibition at 100–200 μM (Fig. 1). This inhibition was largely shifted in the pyridylpiperazine resistant *E. coli* isolate (carrying the S450P AcrB substitution), confirming its resistance to BDM88855.HCl (**9'**). In parallel assays, PAβN inhibition of pyridomycin and oxacillin started from 6.25–12.5 μM and was similar in both the wild-type and pyridylpiperazine resistant *E. coli* isolate. With oxacillin, PAβN was

| Table 2 Minimal Inhibitory concentration (MIC₉₅, all in μg/mL) of a panel of antibiotics on *E. coli* in the presence (+) and absence (−) of 100 μM BDM88855.HCl (9'), as determined by using the resazurin reduction assay. | | | |
|---|---|---|---|
| **Antibiotic** | ***E. coli* BW25113** | | ***E. coli* BW25113 ΔacrAB** |
| | − | + | − |
| Oxacillin | >100 | 3.0 | 1.2 ± 0.37 |
| Linezolid | 100 | 3.1 ± 0.07 | 4.8 ± 2.3 |
| Novobiocin | >100 | 12 | 3.8 ± 1.0 |
| Fusidic acid | >100 | 12 | 10. ± 7.0 |
| Pyridomycin | 9 ± 3 | 0.75 ± 0.25 | 0.62 ± 0.19 |
| Chloramphenicol | 4.5 ± 1.5 | 0.5 | 0.86 ± 0.56 |
| Erythromycin | 18.5 ± 6.5 | 4.0 ± 2.0 | 5.6 ± 1.0 |
| Ciprofloxacin | 0.052 ± 0.014 | 0.013 ± 0.004 | 0.012 ± 0.0047 |
| Piperacillin | 0.39 | 0.10 | 0.24 ± 0.22 |
| Tetracycline | 0.75 ± 0.25 | 0.25 | 0.42 ± 0.18 |
| Triclosan | 0.045 ± 0.005 | 0.026 ± 0.018 | 0.011 ± 0.0023 |
| Ampicillin | 4.5 ± 1.5 | 2.5 ± 0.5 | 3.5 ± 1.4 |
| Ceftazidime | 0.12 | 0.25 | 0.35 ± 0.18 |
| Streptomycin | 3.13 ± 0.005 | 6.3 | 2.8 ± 0.50 |
| Aztreonam | 0.060 | 0.09 ± 0.03 | 0.12 ± 0.032 |
| Gentamicin | 0.59 ± 0.2 | 1.60 | 0.43 ± 0.14 |

As a comparison, the antibiotic susceptibility of *E. coli* lacking *acrAB* is also indicated. BDM88855.HCl (**9'**) on its own has no impact on *E. coli* growth up to the highest concentration tested (250 μM). Data are the mean ± SEM of least two independent biological replicates (SEM is not indicated when all replicates showed the same MIC). Source data are provided as a Source Data file.

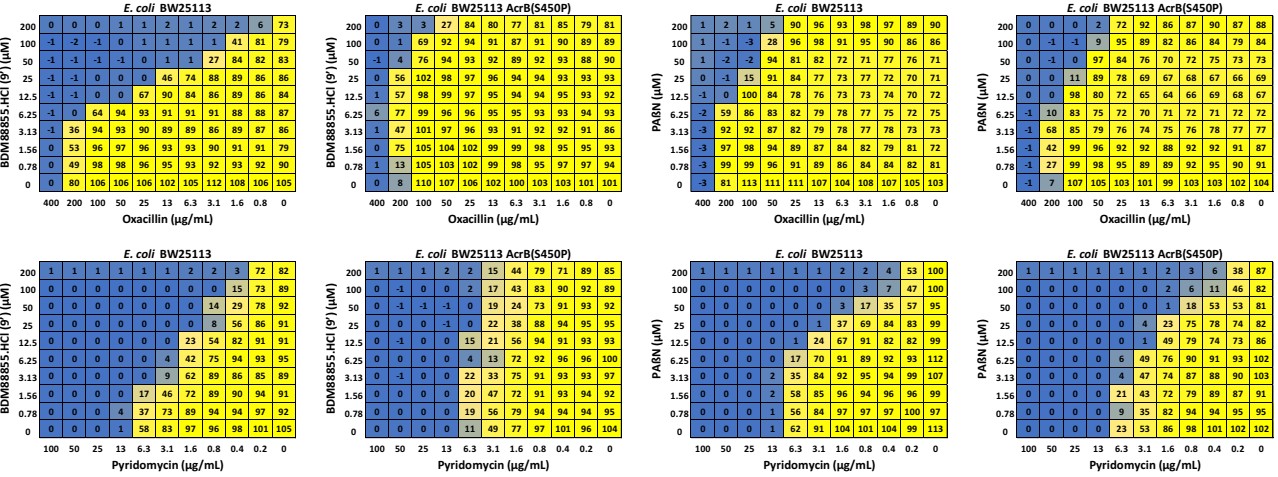

**Fig. 1 Checkerboard assay evaluating pyridomycin and oxacillin antibiotic activity in the presence of different concentrations of BDM88855.HCl (9')** and PAβN. Experiments were performed on both *E. coli* BW25113 and the pyridylpiperazine resistant *E. coli* isolate (carrying the S450P AcrB mutation) to confirm the on-target activity of BDM88855.HCl (**9'**). Bacterial viability was determined using the resazurin reduction assay and is expressed as a percentage resazurin reduction compared to the untreated bacteria, with this viability colour coded from yellow (100%) to blue (0%). Data represent the mean bacterial viability of at least four independent replicates. Source data are provided as a Source Data file.

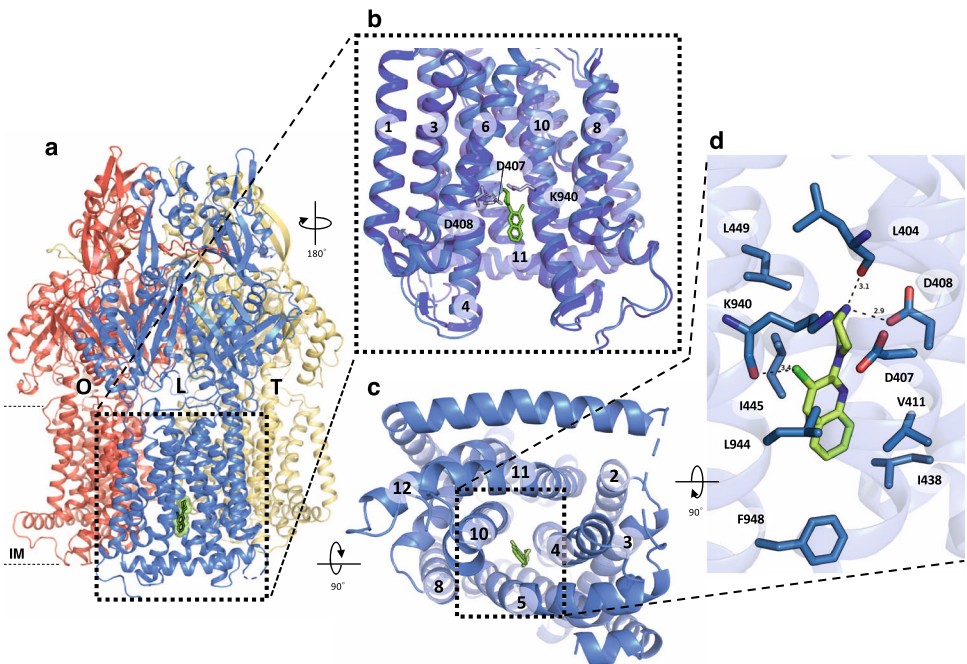

**Fig. 2 Structure of AcrB in complex with BDM88855 (9). a** Side view of the AcrB trimer comprising the L (marine), T (yellow), and O (red) protomers. BDM88855 (**9**) (green) binds to the TM domain of the L protomer. The boundaries of the inner membrane (IM) are indicated for the L protomer. **b** Side view on the TM domain of the L protomer. The apo-structure (dark blue) and the BDM88855 (**9**)-bound L protomers (marine) are superimposed and indicate subtle shifts in helix movements to accommodate the pyridylpiperazine inhibitor. The inward-open TM domain of the AcrB L protomer indicates the binding of BDM88855 (**9**) (green, stick representation) near the proton relay triad D407, D408, and K940. **c** Cytoplasmic side view into the TM domain of the AcrB L protomer (marine) and the location of BDM88855 (**9**) (green, stick representation) nested between TM helices 4, 5, and 10. **d** Enlarged view of the inhibitor binding site showing interacting residues (marine sticks) in less or equal than 4 Å distance from BDM88855 (**9**). The salt bridge between the piperazine ring and D408, the halogen bond between the BDM88855 (**9**) chlorine and the K940 main chain carbonyl oxygen, and the hydrogen-bond between the L404 main chain carbonyl oxygen and the piperazine ring are indicated by dashed lines and numbers represent the distance in Å.

unable to fully revert *E. coli* susceptibility to that of *E. coli* Δ*acrAB* (MIC 0.78 μg/mL), while BDM88855.HCl (**9'**) achieved this.

**Confirmation of direct inhibition of AcrB by pyridylpiperazines**. To reinforce that BDM88855.HCl (**9'**) acts directly on AcrB, its impact was evaluated on the immediate bacterial accumulation of AcrB substrate; berberine, and biophysically, on the stabilization of the inhibitor/AcrB complex using thermal shift assay (TSA). Accordingly, fluorescence uptake assays showed that BDM88855.HCl (**9'**) increased berberine accumulation in *E. coli* expressing wild-type *acrB* in a concentration-dependent manner, while bacteria producing the AcrB variants A446P and S450P were *de facto* unaffected by the inhibitor (Fig. S1). Likewise, TSAs confirmed a concentration-dependent increment of the melting temperature of the purified wild-type AcrB, but not of the AcrB (A446P) variant (Fig. S2), indicating inhibitor-mediated stabilization of the protein/inhibitor complex for wild-type AcrB and not for the resistant mutant. Of note is that the concentration-dependent boosting of antibiotic activity, berberine accumulation and protein stabilization by TSA are congruent. Together, these results reinforce that BDM88855.HCl (**9'**) directly binds and inhibits AcrB.

**Structural confirmation of pyridylpiperazine binding site**. As evidenced by the selection of natural resistant isolates described above, we anticipated inhibitor binding at the TM domain, in contrast to the previously structurally described inhibitors targeting the periplasmic porter domain[19,20]. To confirm this, we determined the co-crystal structure of wildtype AcrB with BDM88855 (**9**) at 2.6 Å resolution, and the inactive AcrB R971A variant[13] with BDM88855 (**9**) and BDM88832 (**8**) at 2.5 and 2.8 Å resolution (Table S8). Non-proteinaceous electron density between TM4, TM5 and TM10 showed that BDM88855 (**9**) and BDM88832 (**8**) bind specifically to the TM domain of the L protomer (Fig. 2, Fig. S3). Analysis of the anomalous signals originating from the iodine atom of BDM88832 (**8**) or the brominated pyridylpiperazine inhibitor (compound **11**) indicated the location of the halogen atoms (Fig. S3) and confirmed the binding of BDM88832 (**8**) and BDM88855 (**9**) at this position in the TM domain, with minimal halogen and hydrogen bond length deviations between the interaction of the latter inhibitor in wildtype and the R971A variant (Fig. S3d).

Superimposition of the unliganded AcrB L protomers[31] (PDB: 4DX5) and inhibitor-bound structures reveals a root mean square deviation (RMSD) ($C_\alpha$) of 0.89–1.05 Å over 1024 residues (Table S9), with the most obvious deviations at the TM domain (Fig. 2b, Table S9). Here, the TM helices show subtle differences in their conformation at the cytoplasmic side, leading to an expansion of the narrow channel between the D407, D408, and K940 residues of the proton relay site and the cytoplasm, thus allowing for the accommodation of BDM88855 (**9**) (Fig. 2b). The quinoline moiety of the inhibitor interacts with surrounding hydrophobic residues, including V411, L442, I445, and F948 (Fig. 2d, Fig. S3a, Table S10). A subtle shift of TM10 permits the formation of a halogen bond[32] between the chlorine atom and carbonyl oxygen of main chain K940. The piperazine moiety is mainly surrounded by residues involved in the proton relay network (D407, D408, and K940), as well as L449, and the carbonyl oxygen of the main chain L404 (Fig. 2d, Fig. S3a). The

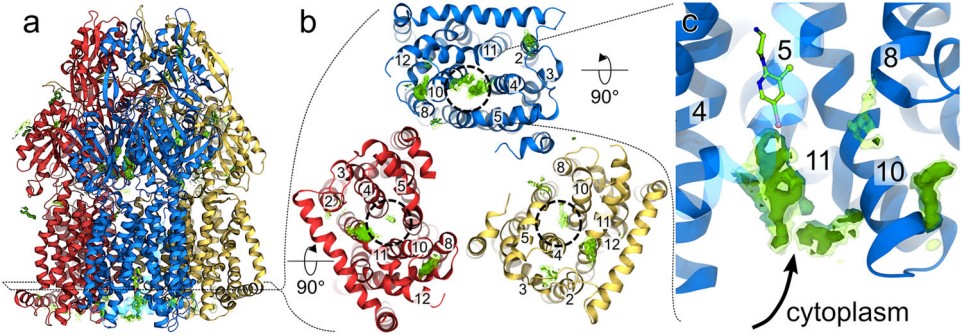

**Fig. 3 Putative entry of BDM88832 (8) to its binding site in AcrB.** Figure shows the preferred cumulative location (green surfaces) of BDM88832 (8) as seen in all-atom MD simulations of several compounds placed around AcrB embedded in a phospholipid bilayer. **a** Side view of BDM88832 (8) density on the whole AcrB tripartite efflux pump (L, T, and O protomers are shown as blue, yellow and red, respectively). **b** BDM88832 (8) density distribution viewed from the cytoplasmic side with only the TM domain shown for clarity. Relevant TM helices are labelled, and dashed circles on each monomer delimit the location of the experimental binding site. **c** Zoomed side view of the L protomer showing accumulation of BDM88832 (8). The X-ray conformation of BDM88832 (8) in AcrB is superimposed in CPK coloured by atom type (C, N, I, and Cl in green, blue, pink, and yellow, respectively). TM helix 5 is shown transparent for clarity. All data are cumulative results from 5 independent MD simulations of 2 μs each; transparent and solid green surfaces represent iso-values of 3 and 5, respectively.

carboxyl moiety of D408 and carbonyl moiety of L404 are re-oriented compared to the apo-structure to allow the formation of a salt bridge with the protonated secondary nitrogen atom ($-NH_2^+-$) of the piperazine moiety (Fig. 2d). Furthermore, the D407 and K940 side chains are re-oriented to accommodate the piperazine moiety of BDM88855 (9), (normally, K940 occupies this area in the unliganded L protomer) (Fig. S4). This re-orientation disrupts the H-bond network between S481, D407, D408, K940, and T978[13], while maintaining H-bonding between S481 and D408 in the BDM88855 (9) co-structure (Fig. S5). The presence of a halogen atom in position 3 of the quinoline ring is required for inhibitor activity and correlates with the formation of a halogen bond to the main chain carbonyl oxygen of K940. In addition, the formation of a salt bridge between the charged piperazine ring and the carboxyl side chain of D408 reinforces the importance of the basic nitrogen of the piperazine ring for activity. Therefore, the inhibitor likely interferes with the proton relay site at the TM domain of AcrB and appears to lock the L protomer in this state.

**Structure-function validation of the BDM88855 binding site.** Drug susceptibility assays show that the AcrB variants I438A, I445A, I943A, and L944A, conferred increased susceptibility against all AcrB substrates tested (Fig. 2d, Table S10, Fig. S6/S7), indicating that these residues may be involved in drug transport and/or proton binding/coupling. AcrB variants V411A, L442A, Glu947A conferred wild-type-like susceptibilities, in the absence and presence of BDM88855 (9) (except linezolid in case of the L442A variant) (Table S10), indicating that these substitutions result in a loss of inhibitor affinity. The alkyl side chains of V411, L442, and Glu947 orient towards the quinoline moiety of the inhibitor resulting in van der Waals interactions (Fig. S3a), so substitution with alanine (with a shorter side chain) most likely leads to a loss of inhibitor interactions and affinity. In contrast to the mutations that prevent AcrB inhibition, the functionally active F948A variant was hypersensitive to inhibition by BDM88855 (9) compared to wildtype AcrB (Table S10, Fig. S7). As F948 resides at the cytoplasmic rim of the TM domain, proximal to the inhibitor binding site, the Ala-substitution may facilitate inhibitor access from the cytoplasm toward its binding site.

**Putative pyridylpiperazine uptake into AcrB.** The location of the key residues at the cytoplasmic rim of the TM domain

(Fig. 3), and the presence of water accessible channels from the cytoplasmic side of this region in the L state[13], suggest that the inhibitor gains access to its binding pocket *via* the cytoplasmic side. To assess this hypothesis, we performed all-atom molecular dynamics (MD) simulations[33] of the interaction between one hundred molecules of BDM88832 (8) and wild-type AcrB (Fig. S9). Computational data confirmed the tendency of one or more inhibitor molecules to persistently reside in a channel linking the cytoplasm and the BDM88832 (8) binding site, as inferred from the crystal structure in the L protomer of AcrB (Fig. 3). A lower density of compounds was seen for both the T and O protomers. BDM88832 (8) interacted mostly with residues V411, L944, E947, and F948 (Fig. S10) as it travelled from the cytoplasm towards the binding pocket. A persistent interaction between the inhibitor and F948 could rationalize the observed hypersensitive F948A phenotype in comparison to wild-type AcrB. Overall, these results validate the AcrB/inhibitor co-structures and provide strong evidence that pyridylpiperazine inhibitors access AcrB from the cytoplasm rather than the periplasm.

## Discussion

In Gram-negative bacteria, RND efflux pumps play a central role in both innate and acquired antibiotic resistance by extruding a plethora of antimicrobial compounds. Therefore, the discovery and development of potent efflux pump inhibitors would allow for improved antibiotic efficacy and drug repurposing[34,35]. In addition, preventing antibiotic efflux would greatly improve the success rate of drug discovery programs, where RND-pump compound extrusion is the origin for low hit rates (New drugs for bad bugs[36]). Some RND-efflux pump inhibitors have already been discovered, including PAβN and derivatives, pyridopyr-imidine (such as D13-9001), and the pyranopyridine derivatives (MBX series). PAβN is a broad spectrum competitive inhibitor of the DBP of RND pumps (but also targets the outer membrane LPS layer[37], and was recently shown to perturb the functional dynamics of AcrB[38]. Both the pyridopyrimidine and pyr-anopyridine inhibitors bind to a specific "hydrophobic trap" in the DBP in the T protomer of asymmetric AcrB (Fig. 4), either preventing substrate binding or inhibiting the T to O transition required for the functional catalytic cycle of the RND pump[19,20]. As the DBP is a lipophilic pocket, medicinal chemistry for optimizing the drug-like properties of hit inhibitor molecules can be challenging. To facilitate downstream medicinal chemistry, in this

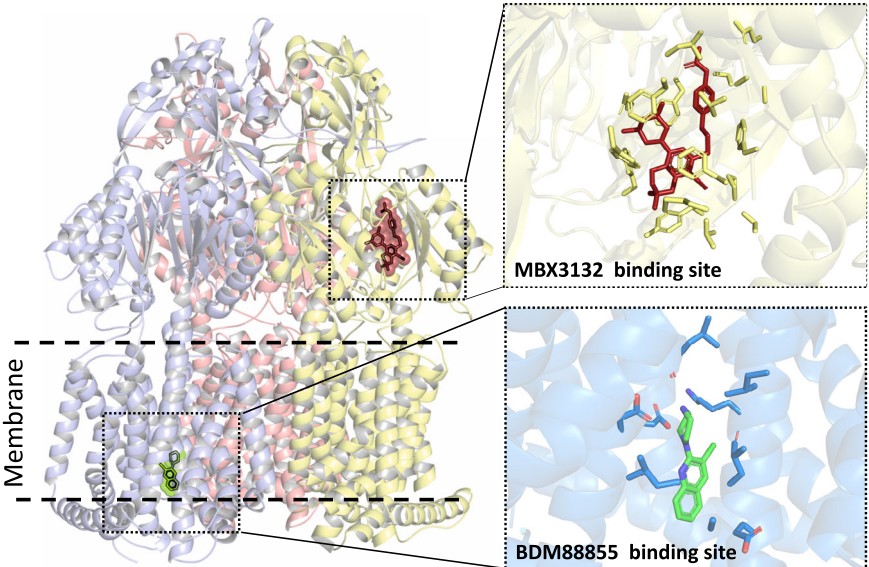

**Fig. 4 Comparison of BDM88855 (9) and MBX3132 binding pocket in AcrB.** Superimposition of the structures of BDM88855 (**9**) (green compound) bound AcrB (pdb 7OUK) and MBX3132 (red compound) bound AcrB (pdb 5ENQ)[19], shows these efflux pump inhibitors to occupy different protomer states of AcrB (the blue T protomer for BDM88855 (**9**) and the yellow L protomer for MBX3132), in spatially distant binding pockets.

work we opted to screen a selected library of small hydrophilic fragments. Due to their physicochemical properties, it was anticipated that more hydrophilic fragments might bind to alternative (non-DBP) sites of AcrB.

A breakthrough in this work was the selection of natural *E. coli* point mutants specifically resistant towards the pyridylpiperazine-based EPIs (at a low frequency). These mutations were located in the AcrB TM domain, suggesting a hitherto undiscovered mechanism of efflux pump inhibition, a finding that was consolidated by structural biology data. Additional mutagenesis data suggest that these EPIs act directly on the AcrAB-TolC efflux pump, rather than on its transcriptional/translational regulation or on other putative targets.

The functional state of the AcrB trimer comprises three protomers in different conformational states, the L, T, and O protomers[15]. From the crystal co-structure, it is apparent that the pyridylpiperazine EPIs bind uniquely to the L protomer of AcrB. MD simulations also predicted pyridylpiperazine inhibitors to preferentially interact with the cytosolic surface of the L protomer. Specifically, simulations indicated that compounds enter AcrB in the L protomer state via a cytoplasmic-open access channel leading to the inhibitor binding pocket. As the L protomer TM domain is in an inward open conformation, this entry pathway is more accessible than in the T or the O protomer, which are in an outward open (inward closed) or occluded conformation, respectively[13].

The elucidated crystal structure of the pyridylpiperazine binding pocket in AcrB points to two possible mechanisms by which these EPIs inhibit AcrB function. Firstly, by binding deep in the TM domain of the L protomer, the pyridylpiperazine may sterically prevent the L to T transition, a vital process involving major structural movement in the TM domain from an inward open to an outward open conformation. An alternative possible mechanism of action is that the pyridylpiperazine impedes AcrB from using the proton motive force as its energy source for substrate translocation. From the identified inhibitor binding pocket, the protonated piperazine moiety of the pyridylpiperazine forms a critical salt bridge with the side chain of D408. D408, together with D407, K940, and R971 are essential catalytic residues of the AcrB proton relay[13,15,24], and the interaction of the pyridylpiperazine with D408 likely interferes with the

H+-translocation pathway. Interestingly, a similar mechanism of action has been proposed for a number of anti-tuberculosis drugs targeting the Gram-positive RND-type mycolic acid efflux pump, MmpL3[39,40]. For this mechanism of action, the pyridylpiperazine would also need to bind to the T protomer of AcrB, as proton binding from the periplasm to D407/D408 occurs during the T to O protomer transition. The reason why no pyridylpiperazine was observed in the T protomer may be of technical nature, where the transition from the L to T protomer may not occur in purified AcrB due to the presence of the stabilizing DARPins. While MD simulations in water solution and model phospholipid membrane also point to preferred access via the L protomer, they were performed in the absence of a proton gradient across the membrane. Future investigations will aim to clarify which of these likely mechanisms leads to AcrB inhibition.

The work presented here characterizes and validates pyridylpiperazine series as a novel class of efflux pump inhibitors that act by binding a previously unexploited pocket in the TM region of the L protomer of AcrB. Our data suggest that occupancy of this conserved pocket prevents the functional catalytic cycle of this RND pump resulting in enhanced drug susceptibility. Together, these lead molecules provide a promising basis for optimization into potent drug-like EPIs for in vivo efficacy and clinical development to disarm Gram-negative bacteria.

## Methods

**Strains, media, and antibiotics**. *Escherichia coli* BW25113 (CGSC 7636) and its derivatives *E. coli* Δ*tolC* (JW5503-1, Δ*tolC732::kan*, CGSC 11430), *E. coli* Δ*acrA* (JW0452-3, Δ*acrA748::kan*, CGSC 11843) and *E. coli* Δ*acrB* (JW0451-2, Δ*acrB747::kan*, CGSC 8609) were obtained from *E. coli* Genetic Stock Center (CGSC, New Haven, Connecticut) and originated from the Keio Collection[41]. Bacterial were regularly cultured on LB broth (BD, DIFCO) at 37 °C. All bacterial antibiotic susceptibility tests were performed in cation-adjusted Mueller-Hinton broth (CAMHB; BD Difco) at 37 °C.

Commercially available molecules, including antibiotics and efflux pump inhibitors, were purchased from various vendors, including Sigma–Aldrich, Carbosynth Limited, Fisher Scientific, Euromedex, and details can be provided on request. Pyridomycin was extracted and purified from *Dactylosporangium fulvum*, as previously described[28].

**Screening library**. To screen for potential EPIs, a chemical library of 1280 compounds (at 100 mM in DMSO) were transferred (in duplicate) by acoustic technology (Echo® 550, Labcyte Inc) to a destination 384-well plate (150 nL/well, the

final concentration of 300 μM). *E. coli* BW25113 was then thawed from frozen stocks and diluted to an OD$_{600}$ of 0.0004 in cation-adjusted Mueller-Hinton broth (CAMHB). This culture was split in two, and spiked with either DMSO (control for booster activity alone), or pyridomycin (final concentration of 5 μg/mL), and transferred to the destination plates containing the chemical library (50 μL per well). Following culture (5 h, 37 °C), *E. coli* viability was evaluated using the resazurin reduction assay, and measured by fluorescence (POLARstar Omega, BMG Labtech: Ex: 530 nm, Em: 590 nm).

**Medicinal chemistry.** The chemical synthesis can be found in the supplementary methods section and structural validation of the compounds can be found in the source data files.

**Liquid MIC determination.** The minimum inhibitory concentration (MIC) determination was performed in 96-well format in order to confirm screening results, define EPI activity on bacterial mutants, and evaluate the activity of improved EPIs. Briefly, bacterial cultures were diluted to OD$_{600}$ of 0.001 in CAMHB and spiked with either EPIs (to determine the shift in antibiotic MIC, typically 500 μM BDM73185 (**1**) or 100 μM BDM88855.HCl (**9'**)) or with AcrAB-TolC substrates (to determine the activity of EPI, typically 8 μg/mL pyridomycin). These bacterial/compound mixes were then transferred to a 96-well plate (100 μL/well), and a dose-response of the compounds of interest was added by serial dilutions. Following culture (5 h, 37 °C), strain viability was evaluated using the resazurin reduction assay, and measured by fluorescence (POLARstar Omega, BMG Labtech: Ex: 530 nm Em: 590 nm). EC$_{90}$ was defined as the compound concentration that allowed a sub-inhibitory dose of pyridomycin (8 μg/mL) to prevent 90% of resazurin turnover compared to the non-treated bacteria.

To screen for the spectrum of antibiotic boosting by BDM88855.HCl (**9'**), antibiotics were added in dose-response by acoustic technology (Echo® 550, Labcyte Inc) to a destination 384-well plate. Wild-type *E. coli* BW25113, *E. coli* resistant clones 1.25_1 (with S450P AcrB mutation) and *E. coli* resistant clones 1.25_2 (with A446P AcrB mutation) were diluted to OD$_{600}$ of 0.001, spiked with DMSO or 100 μM BDM88855.HCl (**9'**) and added to the destination plate (50 μL per well). Viability was determined as described above after 5 h co-culture.

**Cytotoxicity.** The cytotoxicity of compounds BDM73185 (**1**) and BDM88855.HCl (**9'**) was determined on BALB/3T3 cells using live imaging following both Hoechst 33342 and NucView 488 Caspase-3 staining. Briefly, BALB/3T3 cells were seeded in 384-well plate, and 24 h later, compounds BDM73185 (**1**) and BDM88855.HCl (**9'**) were added (0, 12.5, 25, 50, and 100 μM) to the culture medium, as well as Hoechst 33342 and NucView 488 Caspase-3 substrate. 24 h and 48 h after compounds addition, live imaging was performed using an In Cell Analyzer 6000 (GE Healthcare). The cytotoxicity was defined based on the ratio of the apoptotic cell population (NucView staining) and the total population (Hoechst staining) using Columbus software. Compounds were tested in triplicate. Carfilzomib (499, 249,142, 61, and 30.6 nM) was used as a positive control in this assay.

**Isolation of BDM73185 (1)-resistant mutants.** The selection of *E. coli* BDM73185 (**1**)-resistant clones was performed by plating 50 μL of concentrated log-phase BW25113 culture at OD$_{600}$ = 80 onto CAMHB agar containing either erythromycin alone (1.25, 2.5, 5, 10, 20, or 40 μg/mL) or erythromycin with 600 μM BDM73185 (**1**). Plates were incubated at 37 °C until the appearance of resistant colonies (1–2 days). Clones were picked and grown in CAMHB without selective pressure, before confirmation of antibiotic susceptibility (erythromycin, pyridomycin, and linezolid) in the absence and presence of BDM73185 (**1**) at 300 μM.

**gDNA extraction and *acrB* variant analysis.** *E. coli* BW25113 wild-type and the four BDM73185 (**1**)-resistant clones were grown overnight in CAMHB medium at 37 °C. Cells (1 mL) were collected by centrifugation, re-suspended in Phosphate-Buffered saline (500 μL) and were broken using bead beating using the MP Biomedicals™ Instrument FastPrep-24™ (MP Biomedicals, Santa Ana, California). Lysates were collected and incubated with 4.5 μL of 20 mg/mL RNaseA (Pure-Link™ RNase A, ThermoFisher scientific) for 10 min at room temperature. DNA was extracted as per standard phenol-chloroform-isoamyl alcohol (25:24:1) procedure (ThermoFisher scientific). Genomic DNA and purity were checked by DeNovix DS-11 Series (DeNovix, Wilmington, Delaware), and *acrB* sequenced by Sanger sequencing (Genoscreen, France).

**Whole-genome sequencing and variant analysis.** For whole-genome sequencing, genomic DNA from bacteria was submitted to MicrobesNG (http://www.microbesng.com, Birmingham, UK) for library preparation (Nextera protocol) and whole-genome sequencing on an Illumina sequencer (NovaSeq600). Trimmed reads (deposited at NCBI BioProject ID: **PRJNA764862**) were initially processed using PRINSEQ-lite PERL script (**PRINSEQ-lite** version 0.20.4;[42]) to remove low-quality data with the following parameters (-min_len 50 -min_-qual_mean 30 -trim_qual_right 30 -ns_max_n 0 -noniupac). Reads were then aligned to the *E. coli* BW25113 reference genome (NCBI accession number CP009273) with BWA-MEM version 0.7.17-r1188[43]. The output SAM alignment

files were converted to BAM files and sorted using SAMtools version 1.1 view with default parameters[44]. SNPs and In/Dels were called using GATK HaplotypeCaller Version 4.2.0.0[45]. Identified variants were finally annotated using snpEff version 4.3[46].

**Construction of *E. coli* BW25113 *acrB* chromosomal point mutants.** To generate unmarked single point mutant clones, an *E. coli* BW25113 Δ*acrB* mutant was complemented (chromosomal reconstitution) using lambda-Red recombination with either wildtype *acrB*, *acrB* S450P, or *acrB* A446P. Briefly, unmarked BW25113 Δ*acrB* was generated from *E. coli* Δ*acrB* (JW0451-2, Δ*acrB747::kan*,) strain by removal of the FRT-Kan-FRT cassette using the FLP recombinase following transformation with pFLP2[47]. A resulting AmpR KmS mutant was streaked on LB-5% sucrose agar plates to cure the bacteria of pFLP2. This unmarked *E. coli* BW25113 Δ*acrB* strain was then transformed with the inducible Red-recombinase plasmid; pEP1436[48] and AmpR resistant clones were selected at 30 °C. Arabinose induced BW25113 Δ*acrB* (pEP1436) electrocompetent cells were prepared at 30 °C and transformed with three different PCR products containing either wildtype *acrB*, *acrB* S450P, or *acrB* A446P. Specifically, these 3.9 kb PCR products were generated by amplifying *acrB* (primers 5'-ATCACCCTACGCGCTATCTT-3' and 5'- CGCAGCAGGTAAAAGCAGTT-3') from wild-type *E. coli*, or the two selected BDM73185 resistant mutants RC 1.25-1 (with *acrB*(S450P) and RC 1.25-2 (*acrB*(A446P)). As SDS is a good AcrB substrate, recombinants with a complemented functional AcrB were selected by plating on LB agar containing 0.1% SDS and at 37 °C (to also ensure pEP1436 loss). The resulting clones were termed BW25113 Δ*acrB*::acrB(WT), BW25113 Δ*acrB*::acrB(S450P), and BW25113 Δ*acrB*::acrB(A446P), and the correct complementation of *acrB* (and mutants) was confirmed by Sanger Sequencing of *acrB*.

**Whole-cell accumulation assay.** Semi-quantitative accumulation assays were carried out to further characterize the impairment of substrate efflux in the presence of BDM88855.HCl (**9'**). The accumulation of the pump substrate berberine by *E. coli* BW25113 Δ*acrB*/pET24a_*acrB* (WT, D407N, A446P or S450P) leaky expression system was monitored over time in the presence of different BDM88855.HCl (**9'**) concentrations.

Assays were performed similarly to previous experiments[49] with some modifications. In brief, chemically competent *E. coli* BW25113 Δ*acrB* cells were transformed with the corresponding pET24a_*acrB* variants WT, D407N, A446P, or S450P, plated on LB, 50 mg/mL kanamycin (LB-K50) agar and incubated overnight at 30 °C until colonies with a diameter of 1–2 mm were obtained and stored at 4 °C for at least one day. Pre-cultures (3 mL LB-K50 in a 15 mL tube) were inoculated from single colonies and grown at 37 °C under continuous shaking overnight. Each 100 mL LB-K50 were inoculated 1:100 from the pre-cultures and grown in 250 mL smooth-walled glass flasks at 37 °C, 160 rpm until OD$_{600nm}$ reached 0.7–0.9. Cultures were stored on ice for 15 min before 50 mL bacterial cell cultures were harvested (4000 × *g*, 4 °C, 5 min), (i) resuspended and (ii) washed in 1.5 mL potassium phosphate buffer (50 mM potassium phosphate pH 7.5, 1 mM MgSO$_4$) at 4000 × *g*, 4 °C for 3 min. Washed bacteria were resuspended in 750 μL potassium phosphate + 0.2% D-glucose and adjusted to an OD$_{600nm}$ of 2 in an ice-cooled 96-deep well block (within final volumes of 1.6 mL). BDM88855.HCl (**9'**) (6.25 μL of a 25 mM DMSO stock) was dissolved in (243.75 μL) potassium phosphate + 0.2% D-glucose supplemented with 500 μM berberine to 625 μM and serially diluted 1: 5 (50 + 200 μL) to 125, 25, 5, and 1.25 μM. Each 20 μL of the 10-fold stock solutions (and potassium phosphate + 0.2% D-glucose /berberine without BDM88855.HCl (**9'**)) were provided in a 96-black-well plate (Tecan 96 Flat Black, or similar) at RT before 180 μl of the bacteria solutions were quickly added (multichannel dispense mode) to the wells.

Berberine accumulation was monitored spectroscopically at a 96-well fluorescence reader (Tecan Spark) at 28 °C for 40 min with λ$_{ex}$ 365 nm (20 nm bandwidth), λ$_{em}$ 540 nm (20 nm bandwidth, Gain 65), 10 flashes and 40 μs integration time. For further analysis fluorescence values of a certain condition (F$_x$) were normalized on the signal splitting between the inactive D407N variant (F$_{D407N}$) and the wild-type (F$_{WT}$) (both in the absence of the inhibitor) to obtain residual activities (R$_x$): R$_x$ = (F$_x$ − F$_{D407N}$) / (F$_{WT}$ − F$_{D407N}$). Steady-state residual activities (between 1600s and 1800s) from each six independent cultures (*n* = 6) were averaged and plotted against the BDM88855 concentration with the centre of the error bars as the mean value.

**Thermal shift assay (TSA).** Compound induced changes in the AcrB (WT or A446P) melting temperature was measured as a biophysical validation of BDM88855.HCl (**9'**) binding to AcrB, similar to that described previously[50]. Briefly, pET24a_*acrB* variants (WT and A446P) were expressed in *E. coli* C43(DE3) Δ*acrAB*. 4 × 1 L LB medium supplemented with 1 mM MgSO$_4$ and 50 mg/mL kanamycin was inoculated 1:400 with an overnight culture grown from a single colony. Cultures were grown in 2.5 L Tunair shake flasks at 37 °C (135 rpm) until OD$_{600nm}$ reached 0.8–0.9, stored on ice for 30 min, induced with 0.5 mM (final) IPTG and incubated for further 20 h at 20 °C (135 rpm). All following purification steps were performed at 4 °C. Cells were harvested (6000 × *g*, 20 min) and resuspended in 4 volumes (w:v, final) lysis buffer (20 mM TRIS pH 8.0, 500 mM NaCl, 2 mM MgCl$_2$) supplemented with each 10 mg/L lysozyme and DNase for at least

30 min. The protease inhibitor PMSF (200 μM, final) was added directly before cell disruption with a pressure cell homogenizer (Stansted Fluid power Ltd.), 2× times at 2 bar pre-pressure. The lysate was cleared from debris at $20,000 \times g$ for 20 min before subjected to preparative ultra-centrifugation at $142,000 \times g$ for 60 min. Membranes were resuspended in resuspension buffer (20 mM TRIS pH 7.5, 300 mM NaCl, 10% glycerol) to a final concentration of 0.2 mg/mL, snap-frozen in liquid nitrogen and stored at −80 °C. For each protein purification, 4 mL membrane suspensions (800 mg) were thawed and solubilized in 1% DDM (20% w/v stock) and 12 mM imidazole (1 M stock, pH 7.5) in a final volume of 12 mL for at least 30 min under mild sample rotation. Buffer P1 was used for volume adjustment. After ultra-centrifugation ($161,000 \times g$, 30 min) the supernatant was rotated with 400 μL (800 μL 1:1 suspension) HisPur Ni-NTA resin (Thermo Scientific) for at least 30 min. The resin was washed in a gravity flow column with each 15 CV (6 mL) of P1 buffer (20 mM TRIS pH 7.5, 300 mM NaCl, 0.02% DDM) supplemented with (i) 20 mM imidazole and (ii) 80 mM imidazole. The protein was eluted with 4 mL P1, 400 mM imidazole in a further volume of 4 mL P1. Eluates were concentrated in an Amicon Ultra-100 kDa cutoff spin concentrator to 500 μL and applied on a Cytiva Superose 6 increase 10/300 GL size exclusion column with 20 mM HEPES pH 7.5, 150 mM NaCl, 0.02% DDM at a flow rate of 0.25 mL/min. Trimeric protein fractions were collected (at an elution volume of ~13 mL) and diluted to 0.1 mg/mL ($A_{280nm}$) for TSA analysis.

Thermal Shifts Assays (TSA), first described for AcrB by Atzori and colleagues[50] were prepared at room temperature as follows: Volumes of each 0.5 μL (100-fold) inhibitor stock solutions containing 8, 4, 2, 1, 0.5, 0.25, 0.125, 0.0625, or 0 mM (DMSO) BDM88855.HCl (9') were placed in 1.5 mL tubes and diluted (1:100) with 49.5 μL of the protein solution (0.1 mg/mL). Mixtures were incubated for 5 min before aggregates were removed at $13,000 \times g$ for 2 min. After an additional 15 min of incubation 39.5 μL of the supernatant was transferred into fresh 1.5 mL tubes provided with 0.5 μL 1 mg/mL (DMSO) of the cysteine reactive dye CPM (N-[4-(7-diethylamino-4-methyl-3-coumarinyl)phenyl]maleimide). Undissolved material was removed as described before and 30 μL of supernatant were transferred into 250 μL real-time PCR tubes. Samples were heated from 25 °C to 95 °C (1 °C increase every 20 s) while recording CMP-fluorescence ($\lambda_{ex}$ 380 ± 20 nm / $\lambda_{em}$ 460 ± 20 nm) in a Qiagen Rotor-Gene Q PCR machine. For automated data analysis (Rotor-Gene Q software) fluorescence intensities (F) were derived to the temperature (T) and maxima of dF/dT over T were given as melting temperatures ($T_m$). $T_m$ values were calculated from five independent TSA measurements ($n = 5$) with the centre of the error bars as the mean value.

**Site-directed mutagenesis**. Mutagenesis was performed using either ExSite™ (Stratagene) method with 5'-phosphorylated primers or QuikChange™ (Agilent) method. All primers used in this study are listed in (Table S12). All plasmids were verified by sequencing (Genoscreen, France).

**Overproduction and purification of DARPins**. A single colony of E. coli XL1-Blue cells harbouring pQE30-DARPin[10,51] was inoculated in LB liquid medium supplemented with 50 μg/mL kanamycin at 37 °C and cultivated overnight. Overnight cultures were inoculated into a fresh LB liquid medium supplemented with antibiotics as described above. Gene expression was induced with 0.5 mM isopropyl-β-D-thiogalactoside at $OD_{600} = 0.7$ and induced cultures were grown overnight at 37 °C. Cells were harvested by centrifugation and resuspended in 50 mM Tris-HCl buffer at pH 7.5, 400 mM NaCl and 10 mM Imidazole. Cells were lysed by a Pressure Cell Homogeniser (Stansted Fluid Power Ltd, United Kingdom) at 15,000 psi and cleared by centrifugation at $160,000 \times g$ for 1 h. Supernatant was loaded onto a HisTrap HP Ni$^{2+}$ affinity column (5 mL bed volume, GE Healthcare). After two wash steps with the same buffer supplemented with 20 mM and 50 mM of imidazole, DARPin proteins were eluted with 50 mM Tris-HCl buffer at pH 7.5, 400 mM NaCl, 250 mM Imidazole and 10% Glycerol.

**Overproduction and purification of AcrB and AcrB-R971A**. E. coli C43 (DE3) ΔacrAB harbouring pET24acrB$_{His}$[51] (WT or AcrB-R971A variant, respectively) was grown overnight in LB liquid medium supplemented with 50 μg/mL kanamycin at 37 °C. Overnight cultures were inoculated into fresh Terrific Broth liquid medium supplemented with antibiotic as described above and grown until $OD_{600} = 0.8$ before 0.5 mM isopropyl-β-D-thiogalactoside was added to the culture. Cultures were subsequently grown at 20 °C for another 16 h and pelleted by centrifugation. The cell pellet was resuspended in Buffer A (20 mM Tris-HCl at pH 8.0, 500 mM NaCl, 2 mM MgCl$_2$ and 0.2 mM diisopropyl fluorophosphate) and lysed by Pressure Cell Homogeniser (Stansted, United Kingdom). Cell debris was removed by centrifugation at $23,000 \times g$ for 15 min and cell membranes were collected by centrifugation at $160,000 \times g$ for 2 h. Cell membranes were resuspended in Buffer B [20 mM Tris/HCl buffer at pH 7.5, 150 mM NaCl, 20 mM Imidazole and 10% Glycerol and solubilized with 1% dodecyl maltoside (D-97002-C, DDM, Glycon)] at 4 °C for 1 h. Solubilized membranes were cleared at $160,000 \times g$ for 30 min and the supernatant was loaded onto a HisTrap HP Ni$^{2+}$ affinity column (1 mL bed volume, GE Healthcare). The column was washed with Buffer B supplemented with 0.02% DDM in addition of 60 mM and 90 mM imidazole, respectively. AcrB protein was eluted with Buffer D (20 mM Tris/HCl at pH 7.5, 150 mM NaCl, 220 mM Imidazole, 10% Glycerol and 0.02% DDM).

**Co-crystallization of AcrB/DARPins with BDM88832 (8) and BDM88855 (9)**. Before setting up crystallization, AcrB was mixed with DARPins in a molar ratio of 1:2 and incubated on ice for 15 min. Excess DARPins were removed by size-exclusion chromatography (Superose 6, GE Healthcare) with buffer containing 20 mM Tris pH 7.5, 150 mM NaCl, 0.03% DDM and 0.05% DDAO (Anatrace). Co-crystallization of AcrB/DARPins variants with various inhibitors were obtained by hanging drop crystallization within 1–2 weeks. Briefly, BDM88832 (8) or BDM88855 (9) (6–8 mM) were mixed with a protein solution (12 mg/mL final concentration), and incubated on ice for 30 min. Subsequently, all the protein solutions were centrifuged at $13,000 \times g$ for 10 min at 4 °C to remove the insoluble materials, before setting up the crystallization. The reservoir contained 50 mM ADA pH 6.6, 5% Glycerol, 8–9% PEG4000 and 110–220 mM ammonium sulfate. Crystals were harvested after 2 months and cryo-protected by serial transfer of crystals into reservoir supplemented with increasing of glycerol concentration to 28% before flash-cooling in liquid nitrogen.

**Diffraction data collection and refinement**. Data were collected on beamline PROXIMA 2 A (Soleil Synchrotron, Saint Aubin, France) using an Eiger detector (Dectris), respectively, indexed and integrated with XDS[52]. All the structural models were iteratively built-in COOT[53] with a structural model (PDB ID: 5JMN)[54] and refined with REFMAC5 in CCP4i v7[55]. The structure was validated with MolProbity v4.5[56]. Polder electron density maps were calculated by phenix.polder[57]. All figures were generated by Pymol (Schrödinger LLC).

**Drug agar plate assay**. A colony of E. coli BW25113 ΔacrB harbouring pET24acrB-His ecoding wild-type (WT) or variant AcrB were grown overnight in LB medium containing 50 μg/mL kanamycin at 37 °C. Dilution of the cultures to $OD_{600}$ $10^{-1}$–$10^{-6}$ was prepared and 4 μL of each diluted culture were spotted on LB agar plates containing 50 μg/mL kanamycin, supplemented with AcrB drugs (12.5 μg/mL dicloxacillin, 12.5 μg/mL fusidic acid, 12 μg/mL linezolid, or 80 μg/mL TPP$^+$). Plates were incubated at 37 °C for 14–16 h. Agar plates were imaged by ImageQuant TL (GE Healthcare BioSciences AB, Uppsala, Sweden). The intensity of the cell growth (Table S10) was quantified by ImageJ 1.52o software. The cell growth of WT at first dilution is set to 1. The intensity of cell growth at each dilution higher than 10% of cell growth of WT at first dilution is considered as cell growth. To calculate the relative cell growth of each mutant, the cell growth of each mutant is normalized with cell growth of WT after deduction of the cell growth of cells harbouring the inactive AcrB D407N variant (cell growth = 0). E.g. if the cell growth of cells harbouring AcrB WT is 4 dilution steps, the fourth dilution step is set to 100% growth). Growth of cells harbouring AcrB variants lower than 75% of the growth of cells containing AcrB WT is considered significantly affected if $p < 0.005$ or slightly affected if $p < 0.05$. Growth of cells harbouring AcrB variants in between 75–85% of growth of cells harbouring AcrB-WT is considered as slightly affected if $p < 0.005$. p value was calculated by two-sided Student's t-test (Table S10).

**Western blot analysis**. Western blot analysis of the production of AcrB protein was performed as previously described[54]. Briefly, 3 μL of overnight cultures with $OD_{600} = 1.0$ were suspended in 18 μL of Lysis Buffer (50 mM Tris-HCl, pH7.5, 500 mM NaCl, 10% Glycerol) and incubated with 2× SDS-lysis buffer. The cell suspensions were incubated at 95 °C for 10 min. The cell lysates were centrifuged at $13,000 \times g$ for 10 min. The supernatant was resolved by 12.5% SDS–PAGE gels and transferred onto nitrocellulose membrane. The membrane was incubated with primary rabbit anti-AcrB antibody (dilution of 1:10,000; Neosystems, France, custom-antibody) and then, with a secondary goat anti-rabbit IgG (whole molecule)-alkaline phosphatase antibody (dilution of 1:1,500; A3687, Sigma–Aldrich, St. Louis, USA). Finally, the blot development was performed with NBT (nitro-blue tetrazolium chloride) and BCIP (5-bromo-4-chloro-3'-indolyphosphate p-toluidine salt). Unmodified western blots are shown in Fig. S8.

**Molecular dynamics simulations**. All-atom molecular dynamics (MD) simulations of AcrB embedded in a model phospholipid bilayer and a 0.1 M aqueous NaCl solution containing 100 molecules of BDM88832 (8) were performed with the AMBER18 package[58]. 100 molecules were used to increase the probability to detect uptake events. Protomer-specific protonation states were adopted for AcrB following the previous works[13,59,60]: residues E346 and D924 were protonated only in L and T protomers, while residues D407, D408, and D566 were protonated only in the O protomer. The topology and the initial coordinate files (the experimental structure of AcrB with PDB ID 4DX7 was used as starting conformation) were created using the LEaP module of the AMBER18 package. The protein was embedded in a mixed bilayer patch composed of 1-palmitoyl-2-oleoyl-sn-glycero-3-phosphoethanolamine (POPE) and 1-palmitoyl-2-oleoyl-sn-glycero-3-phosphoglycerol (POPG) in a 2:1 ratio, for a total of ~650 lipid molecules symmetrically distributed across the two bilayer leaflets. The AMBER force field protein.ff14SB[61] was used to represent the protein, the lipid17 (http://ambermd.org/GetAmber.php) parameters were used for the phospholipids, the TIP3P model was employed for water[62], and ions parameters were taken from ref. [61]. The General Amber Force-Field (GAFF)[63] was used to parametrize BDM88832 (8), using the antechamber

tool of AMBER and atomic charges derived from an electrostatic potential distribution calculated with Gaussian16[64] as in ref. [65]. The systems were then heated from 0 to 310 K in two steps: (i) from 0 to 100 K in 1 ns under constant-volume conditions, with harmonic restraints ($k = 1\ kcal\cdot mol^{-1}\cdot\text{Å}^{-2}$) applied to the heavy atoms of both the protein and the lipids; (ii) from 100 to 310 K in 5 ns under constant pressure (set to a value of 1 atm) with restraints on the heavy atoms of the protein and on the z coordinates of the phosphorous atoms of the lipids, so as to allow membrane rearrangement during heating. As a final step before production runs, a series of 20 equilibration steps, each of 500 ps in duration (total 10 ns) and with restraints on the protein coordinates, were performed to equilibrate the simulation. These equilibration steps were carried out under isotropic pressure scaling using the Berendsen barostat, whereas a Langevin thermostat (with a collision frequency of 1 ps$^{-1}$) was used to maintain a constant temperature. Finally, five MD simulations of 2 μs each were performed under an isothermal-isobaric ensemble. A time step of 2 fs was used for all runs before production, while the latter runs were carried out with a time step of 4 fs after hydrogen mass repartitioning[69]. R–H bonds lengths were constrained using the SHAKE algorithm, and coordinates were saved every 500 ps. The Particle mesh Ewald algorithm was used to evaluate long-range electrostatic forces with a non-bonded cut-off of 9 Å.

**Post-processing of the MD trajectories**. MD trajectories were analyzed using either in-house *tcl* and *bash* scripts or the *cpptraj* tool of AMBER18. Hydration of the TM region of the protein (including the residues lining the binding sites of BDM88832 (**8**) and BDM88855 (**9**)) was estimated by calculating the average density values along the cumulative trajectory, using the *grid* tool of AMBER18. Densities were normalized to the value of bulk water (1 g/mL). Figures were prepared using xmgrace (https://plasma-gate.weizmann.ac.il/Grace/) and VMD 1.9.4 (23).

**Reporting summary**. Further information on research design is available in the Nature Research Reporting Summary linked to this article.

## Data availability

Atomic coordinates and structure factors reported in this paper have been deposited in the Protein Data Bank under accession numbers, 7OUK (BDM88855 inhibitor bound to the transmembrane domain of AcrB), 7OUL (BDM88832 inhibitor bound to the transmembrane domain of AcrB-R971A), 7OUM (BDM88855 inhibitor bound to the transmembrane domain of AcrB-R971A). Atomic coordinates that were used and support the findings of this study are available in the Protein Data Bank under accession numbers 4DX5, 4DX7, 4U96, and 5JMN. Whole-genome sequencing data (fastq files) for parental and BDM73185 resistant *E. coli* isolates have been deposited at NCBI (BioProject ID: PRJNA764862). Source data are provided with this paper.

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

## Acknowledgements

We thank Rudy Antoine for bioinformatics support to confirm genetic variants from whole-genome sequencing data, Valérie Landry for cytotoxicity experiments, Virginie Meurillon and Andrea Bosin for technical assistance and Paolo Ruggerone for useful suggestions and fruitful discussions. This research was financially supported by l'Agence Nationale de la Recherche (ANR, France) in partnership with the Bundesministerium für Bildung und Forschung (BMBF, Germany) (program EFFORT, ANR-19-AMRB-0007 (RCH, MF), BMBF- 16GW0236K (KMP)). GM and AVV received support from the National Institutes of Allergy and Infectious Diseases project number AI136799. Research was further supported by Feder (12001407 (D-AL) Equipex Imaginex BioMed), ATIP-Avenir, Institut National de la Santé et de la Recherche Médicale, Centre National de la Recherche Scientifique, Université de Lille, Institut Pasteur de Lille, Région Hauts-de-France. The NMR facilities were funded by the Région Nord-Pas de Calais (France), the Ministère de la Jeunesse, de l'Education Nationale et de la Recherche (MJENR), the Fonds Européens de Développement Régional (FEDER) and the Université de Lille. This research used the Savio computational cluster resource provided by the Berkeley Research Computing program at the University of California, Berkeley (supported by the UC Berkeley Chancellor, Vice Chancellor for Research, and Chief Information Officer). We acknowledge SOLEIL Synchrotron in Saint Aubin, France, for provision of synchrotron radiation facilities and we would like to thank Dr. William Shepard and Dr. Serena Sirigu for assistance in using beamline Proxima 2 A (Proposal Number: 20190173 and 20180035). The synchrotron trips to SOLEIL Synchrotron were partly supported by iNEXT, under PID: 7108, funded by the Horizon 2020 programme of the European Union.

## Author contributions

C.P., H.K.T., A.V.D.C., N.C., J.C.J.C., W.E.F., G.M., E.P., A.V.V., R.T.M., K.M.P., M.F., and R.C.H. designed experiments. C.P. and Ad.H. performed EPI screening and C.P. performed selection and characterization of resistant isolates. N.W. and M.F. designed the compounds and B.D. advised on compounds design. A.V.D.C., N.C., A.B., and P.M. performed the chemical synthesis. C.P. and E.P. generated genetic knockouts. J.C.J.C. performed checkerboard assays. An.H. performed protein production, membrane preparation, and western blot experiments. H.K.T., W.E.F., M.A.K., and An.H. performed the protein purification, crystallization, data collection, and biochemical/microbiological assay. H.K.T., W.E.F., and K.M.P. determined crystal structures. R.T.M. conducted berberine accumulation and thermal shift assays. G.M. contributed parameters of BDM compounds. A.V.V. performed MD simulations and analysed simulation data. All authors contributed to data analysis. C.P., H.K.T., A.V.D.C., N.C., W.E.F., N.W., G.M., A.V.V., K.M.P., M.F., and R.C.H. wrote the paper. K.M.P., M.F., and R.C.H. obtained funding for this work.

## Competing interests

C.P., H.K.T., A.V.D.C., N.C., J.C.J.C., R.T.M., K.M.P., M.F., N.W., and R.C.H. are inventors on patent application covering the EPI described in this manuscript. The remaining authors declare no competing interests.
