## [Peer Review File · Nature Communications]

Pyridylpiperazine-based allosteric inhibitors of RND-type multidrug efflux pumpsREVIEWER COMMENTS

Reviewer #1 (Remarks to the Author):

The work presented in the manuscript characterizes and validates pyridylpiperazine based molecules as a novel class of efflux pump inhibitors that act by binding a previously unexploited pocket in the TM region of the L protomer of AcrB. Further data suggest that occupancy of this conserved pocket prevents the functional catalytic cycle of this RND pump and leads to its inability to efflux a plethora of antibiotic substrates resulting in enhanced drug susceptibility. However, all the experiments in the manuscript did not prove that the pyridylpiperazine compounds acted specifically on AcrB. Therefore, the following experiments should be supplemented before considering publication.

1. The effect of the compounds on substrate transport
2. The effect of the compounds on the bacterial outer membrane
3. The effect of the compounds on the bacterial inner membrane
4. In vitro cytotoxicity toward mammalian cells.

Reviewer #2 (Remarks to the Author):

In the manuscript entitled "A new and potent family of allosteric inhibitors of RND-type multidrug efflux pumps", the authors describe their efforts to discover a novel chemical scaffold that is able to boost antibiotic activity in *E. coli* by targeting the proton motive force-driven inner membrane protein AcrB to inhibit the AcrAB-TolC efflux pump. What's more exciting is that compared with previous studies, the compound pyridylpiperazine acts by binding to the transmembrane (TM) region of AcrB L protomer. This is another major breakthrough for the drug discovery of RND family following the findings that the inhibitors target to the proton translocation channel in the Gram-positive RND-type mycolic acids transporter MmpL3. This study comprehensively uses varieties of techniques and methods, and displays an entire process from inhibitor discovery, target identification, to the studies of the mechanism of action. The research content and results are impressive. This is a very interesting work that would make an important contribution to the field.

I recommend that Nature Communications accept this article with the following suggestions.

1. Although the complex structures of multiple inhibitors and AcrB have been determined, it may be necessary to determine the binding affinity of these inhibitors with AcrB, which may help to better understand the structure information and cell-level activity data.
2. Although the authors explained the possible influences of the resistance mutations (I438A, I445A, I943A, and L944A) for inhibitor binding based on the structural information, the affinity of pyridylpiperazine in combination with mutagenesis and or computational approaches are required to elucidate the molecular basis.
3. The author solved the co-crystal structures of the wild-type and mutant (R971A) AcrB with BDM8855 (9). In figure S1, we find that the binding modes of BDM8855 and AcrB is slightly different in two structures. Therefore, it is necessary to add the structural comparison for them, especially the inhibitor binding pocket, and also need to explain the difference in the main text.
4. Although homotrimeric AcrB comprises three protomers with different conformational states known as loose (L), tight (T) and open (O), the author should add the comparison of the three protomers after inhibitor binding, which may be give a good explanation why the inhibitors inclined to bind to the AcrB L protomer.
5. Why choosing the complex structure of AcrB and BDM88832 (8) to simulate how the compound enters to the TM region of AcrB? Could the structure of AcrB and BDM8855 (9) get the same conclusion?

Reviewer #3 (Remarks to the Author):

New antimicrobials are urgently needed to treat infectious arising from multidrug resistant strains of bacteria. In this study, the authors have identified and characterised a novel allosteric inhibitor of the AcrAB-TolC efflux pump that occupies a site outside of the canonical substrate binding pocket in a transmembrane region not previously identified as an inhibitor target. A screening procedure was used to identify resistant mutants that are located in the pump. The author chemically optimised the lead to generate a potent inhibitor of the AcrB transporter. The results are impressive and are based on extensive quantitative analysis as well as structural biology approaches to confirm the binding site of the inhibitor. The manuscript is well written and reports findings that are significant and of interest to specialised and non-specialised audiences alike. There are a few points below that will hopefully be helpful for the authors of the manuscript to consider.

1) Section "spontaneous AcrB point mutations confer BDM73185 resistance".

The spontaneous AcrB mutants were found with targeted Sanger sequencing, but there could be mutations elsewhere that confound the phenotype. In exploring these mutants, did the authors try complementing acrb null strain with the mutant on a plasmid for BDM73185 resistance?

2) In a related point, the BDM73185 resistant strains have been selected using erythromycin and the susceptibility has been investigated with both erythromycin and pyridomycin. But knowing that erythromycin is also a substrate of the MacAB-TolC efflux pump and that, pyridomycin could be exported by another efflux pump(s) besides AcrAB-TolC, how could the authors be confident that the lack of changes with both antibiotics confirms that there is no change AcrAB-TolC efflux pump basal activity?

3) Line 105, pyridomycin is introduced as a good substrate of AcrAB-TolC efflux pump. Is there evidence for this, as pyridomycin has been described as an anti-tuberculosis drug. Is pyridomycin exported exclusively by AcrAB-TolC pump? Or is there any other pump that could be involved?

4) In the supplementary information method section screening library and liquid MIC determination, is there a particular reason the authors chose to dilute the E. coli BW25113 stock to OD600 of 0.0004 and 0.001?

5) Figure S7 does not appear to be included properly in the document – the figure legend describes a cartoon representation of AcrB not illustrated here.

6) The quality of Figure S5 could be improved.

7) Lines 227-230: It would be helpful to explain from the structural analysis how these residues are important in proton binding/coupling and how the inhibitor interferes with this process.

Does the BDM88855 compound have any NMR spectral features that might undergo chemical shift changes when bound to AcrB? If it does, this might be exploited in the future for NMR analysis of in vivo binding.

Minor points;

Line 48 "as a critical priority"

Line 86 Image of this and how it compares to the other compounds mentioned in the above paragraph might be helpful, just to give a visual guide what EPI look like.

Line 116 It doesn't appear to mediate boosting in Δ acrA either - why not mention that?

Line 115 why were both BDM88855 and BDM88855.HCl tested?

Line 248 Difficult to see green rectangle with blue dots as they are so small in the figure.

Table 1 might be subdivided a bit more clearly e.g. different background colour for 1-7, 8, 9 and 9'-12. The naming is a bit unclear. Why are 2-7 just unnamed variants of BDM73185 but 8 is BDM88832? Was this a pre-existing name for it? Or one given later as it appeared to be a good EPI?

Table S2 Confidence limits would be very helpful to determine the significance of these changes.

Why was BDM73185 tested against a smaller range of antibiotics compared to BDM88855.HCl?

[Please see also an attached PDF file with suggested changes to improve the presentation]

RESPONSE TO REVIEWER COMMENTS

Nature Communications manuscript NCOMMS-21-26975-T

Revision of "A new and potent family of allosteric inhibitors of RND-type multidrug efflux pumps" for Nature communications

Reviewer #1 (Remarks to the Author):

The work presented in the manuscript characterizes and validates pyridylpiperazine based molecules as a novel class of efflux pump inhibitors that act by binding a previously unexploited pocket in the TM region of the L protomer of AcrB. Further data suggest that occupancy of this conserved pocket prevents the functional catalytic cycle of this RND pump and leads to its inability to efflux a plethora of antibiotic substrates resulting in enhanced drug susceptibility. However, all the experiments in the manuscript did not prove that the pyridylpiperazine compounds acted specifically on AcrB. Therefore, the following experiments should be supplemented before considering publication.

1. The effect of the compounds on substrate transport
2. The effect of the compounds on the bacterial outer membrane
3. The effect of the compounds on the bacterial inner membrane

Response: We thank the reviewer for the helpful suggestions. Our understanding of points 1-3, is that the reviewer requests further validation that the EPI act directly on AcrB, rather than indirectly through modulation of AcrB expression or changes in the inner or outer membrane. To a large extent these questions were already addressed in the submitted version of the paper, where we clearly showed that the impact of the EPI is dependent on AcrB (no impact in knockout strains), and this boosting effect is not observed in the presence of the isolated (and reintroduced) EPI-resistant single point acrB mutants. To further support this, the authors have performed 2 additional experiments that demonstrate that the EPI act immediately on the accumulation of a fluorescent substrate (berberine), and that this is not seen when the AcrB A446P or S450P variant is expressed. In addition, biophysical, thermal shift assays have been performed that confirm the concentration dependent stabilisation of AcrB by BDM88855, but not the AcrB mutant (A446P). Both these additional validation assays show a concentration dependent activity of the inhibitor directly in line with the boosting observed for antibiotic activity. Together, these additional experiments further validate that BDM88855 acts directly on AcrB.

Changes made:

Insertion of paragraph into main text describing results:

Confirmation of direct inhibition of AcrB by BDM88855

To reinforce that BDM88855.HCl (9') acts directly on AcrB, its impact was evaluated on the immediate bacterial accumulation of AcrB substrate; berberine, and biophysically, on the stabilization of the inhibitor/AcrB complex using thermal shift assay (TSA). Accordingly, fluorescence uptake assays showed that BDM88855.HCl (9') increased berberine accumulation in E. coli expressing wild-type acrB in a concentration dependent manner, while bacteria producing the AcrB variants A446P and S450P were de facto unaffected by the inhibitor (Fig. S1). Likewise, TSA assays confirmed a concentration-dependent increment of the melting temperature of the purified wildtype AcrB, but not of the AcrB (A446P) variant (Fig. S2), indicating inhibitor-mediated stabilisation of the protein/inhibitor complex for wild-type AcrB and not the resistant mutant. Of note is that

the concentration- dependent boosting of antibiotic activity, berberine accumulation and protein stabilisation by TSA are congruent. Together these results reinforce that BDM88855.HCl (9') directly binds and inhibits AcrB.

Insertion of Materials and methods into supplementary information.

Insertion of Data into supplementary figure (**fig S1 and S2**).

Insertion of source data.

4. In vitro cytotoxicity toward mammalian cells.

Response: The cytotoxicity of BDM73185 (1) and BDM88855.HCl (9') was determined on BALB/3T3 cells, and the compounds were found non-cytotoxic at the maximum concentration tested (100 μ M)

Changes made:

Insertion of paragraph into main text describing results:

Both BDM73185 (1) and BDM88855.HCl (9') did not induce apoptosis (CC50 > 100 μ M) in cytotoxicity assays on BALB/3T3 cells following 24 and 48 hours of exposure and were therefore considered non-cytotoxic in vitro.

Insertion of Materials and methods into supplementary information.

Reviewer #2 (Remarks to the Author):

In the manuscript entitled “A new and potent family of allosteric inhibitors of RND-type multidrug efflux pumps”, the authors describe their efforts to discover a novel chemical scaffold that is able to boost antibiotic activity in *E. coli* by targeting the proton motive force-driven inner membrane protein AcrB to inhibit the AcrAB-TolC efflux pump. What’s more exciting is that compared with previous studies, the compound pyridylpiperazine acts by binding to the transmembrane (TM) region of AcrB L protomer. This is another major breakthrough for the drug discovery of RND family following the findings that the inhibitors target to the proton translocation channel in the Gram-positive RND-type mycolic acids transporter MmpL3. This study comprehensively uses varieties of techniques and methods, and displays an entire process from inhibitor discovery, target identification, to the studies of the mechanism of action. The research content and results are impressive. This is a very interesting work that would make an important contribution to the field. I recommend that Nature Communications accept this article with the following suggestions.

1. Although the complex structures of multiple inhibitors and AcrB have been determined, it may be necessary to determine the binding affinity of these inhibitors with AcrB, which may help to better understand the structure information and cell-level activity data.

Response: Thank you for this valuable suggestion. To help define the biophysical interaction of AcrB with the efflux pump inhibitors, thermal shift assays have been performed that clearly show a concentration dependent stabilisation of the AcrB/inhibitor complex, that is absent for the AcrB(A446P) variant. The concentration-dependent stabilisation of the AcrB-inhibitor complex is aligned to the boosting of berberine accumulation and antibiotic activity. This data clearly confirms a physical interaction of inhibitor with AcrB.

For changes in the manuscript, see the insertion of the paragraph into main text describing results above.

2. Although the authors explained the possible influences of the resistance mutations (I438A, I445A, I943A, and L944A) for inhibitor binding based on the structural information, the affinity of pyridylpiperazine in combination with mutagenesis and or computational approaches are required to elucidate the molecular basis.

Response: Thank you for this suggestion. The data presented show that AcrB variants I438A, I445A, I943A, and L944A are inactive or heavily impaired. It may be of interest to understanding how these substitutions inactivate the AcrB pump, however, this is beyond the scope of the current manuscript, where the focus is on those substitutions that do not impact AcrB function, but only affect the inhibition by the efflux pump inhibitor.

3. The author solved the co-crystal structures of the wild-type and mutant (R971A) AcrB with BDM8855 (9). In figure S1, we find that the binding modes of BDM8855 and AcrB is slightly different in two structures. Therefore, it is necessary to add the structural comparison for them, especially the inhibitor binding pocket, and also need to explain the difference in the main text.

Response: The difference between the structures is minimal. This has been clarified in the text and a structural comparison (new supplementary Figure S3d) has been added to the supplementary data.

Changes made.

To the paragraph on “Structural confirmation of pyridylpiperazine binding site” the following sentence has been added:

“...with minimal halogen and hydrogen bond length deviations between the interaction of the latter inhibitor in wildtype and the R971A variant (Fig S3d).”

To Fig S1 (which is now Fig S3), a fourth panel has been added (S3d) showing the requested structural comparison.

4. Although homotrimeric AcrB comprises three protomers with different conformational states known as loose (L), tight (T) and open (O), the author should add the comparison of the three protomers after inhibitor binding, which may give a good explanation why the inhibitors inclined to bind to the AcrB L protomer.

Response: We thank the reviewer for this suggestion. We interpret this comment as a suggestion to make a superposition of the EPI binding site between the unbound T and O protomer with the BDM88855 (9) bound L-protomer, to assess a possible steric clash. As shown in Figure S4, superposition of the BDM88855 (9) bound L-protomer with the apo L-protomer already demonstrates a steric clash between BDM and the apo-L protomer, suggesting that conformational adaptation is needed to accommodate the inhibitor. For this reason, performing similar superposition with the T and O protomer is of limited value as one cannot anticipate conformational adaptation upon BDM88855 (9) binding to T and O.

5. Why choosing the complex structure of AcrB and BDM88832 (8) to simulate how the compound enters to the TM region of AcrB? Could the structure of AcrB and BDM8855 (9) get the same conclusion?

Response: We thank the reviewer for this note. We could have used either of the two compounds for which the experimental structure of the complex with AcrB is available to simulate inhibitor uptake into the binding pocket. However, due to the demanding calculations needed to investigate this process thoroughly (that is, multiple and long all-atom MD simulations of a large system), we decided to choose one compound. Given the role of F948 as probable gating residue between the cytoplasmic entrance and the binding site, we opted for BDM88832 (8), as the two adjacent rings present in BDM88855 (9) could make more persistent interactions with F948, further limiting the sampling of uptake dynamics. Nonetheless, we believe that similar simulations of the uptake dynamics of compound BDM88855 would have led to the same conclusions drawn for compound BDM88832.

Reviewer #3 (Remarks to the Author):

New antimicrobials are urgently needed to treat infectious arising from multidrug resistant strains of bacteria. In this study, the authors have identified and characterised a novel allosteric inhibitor of the AcrAB-TolC efflux pump that occupies a site outside of the canonical substrate binding pocket in a transmembrane region not previously identified as an inhibitor target. A screening procedure was used to identify resistant mutants that are located in the pump. The author chemically optimised the lead to generate a potent inhibitor of the AcrB transporter. The results are impressive and are based on extensive quantitative analysis as well as structural biology approaches to confirm the binding site of the inhibitor. The manuscript is well written and reports findings that are significant and of interest to specialised and non-specialised audiences alike. There are a few points below that will hopefully be helpful for the authors of the manuscript to consider.

1) Section "spontaneous AcrB point mutations confer BDM73185 resistance". The spontaneous AcrB mutants were found with targeted Sanger sequencing, but there could be mutations elsewhere that confound the phenotype. In exploring these mutants, did the authors try complementing *acrB* null strain with the mutant on a plasmid for BDM73185 resistance?

Response: Thank you for this valuable suggestion. To address this point, 2 sets of additional experiments were performed. 1), whole genome sequencing of the parental *E. coli* strain and BDM73185 resistant mutants was performed, which confirmed that the mutations identified in *acrB* are the only mutations in the whole genome. 2) Starting from an *E. coli* Δ *acrB* mutant, reverse genetics was used to introduce by recombineering into the original chromosomal location, either a wild-type or mutated copy of *acrB*. The resulting engineered mutants were found to be resistant to BDM73185 (1) and BDM88855.HCl (9’).

Together these data confirm indisputability that the identified mutations in *acrB* are unique in the bacteria and cause the phenotypic resistance observed to the EPI, pointing to AcrB inhibition as the predominant mechanism by which the antibiotic activity is boosted.

Changes made.

To the paragraph on “Spontaneous AcrB point mutations confer BDM73185 resistance”, the following sentences has been added:

*“Whole genome sequencing and variant analysis confirmed that no other mutations were selected in these BDM73185 (1) resistant mutants, implying that the resistance phenotype selected was linked solely to an *acrB* point mutation.”*

And at the end of the paragraph

*“As an additional validation, a reverse genetics approach by recombineering was used to specifically reintroduce wild-type *acrB*, *acrB* t1348c [S450P] or *acrB* g1336c [A446P] into its natural chromosomal locus in *E. coli* Δ *acrB*, and these engineered strains showed the same BDM73185 (1) resistance phenotype as the selected strains (Table S6). Together these data imply that the identified *acrB* mutations are solely responsible for the BDM73185 (1) resistance phenotype.”*

2) In a related point, the BDM73185 resistant strains have been selected using erythromycin and the susceptibility has been investigated with both erythromycin and pyridomycin. But knowing that erythromycin is also a substrate of the MacAB-TolC efflux pump and that, pyridomycin could be exported by another efflux pump(s) besides AcrAB-TolC, how could the authors be confident that the lack of changes with both antibiotics confirms that there is no change AcrAB-TolC efflux pump basal activity?

Response: Thank you for this insightful question. The basal antibiotic susceptibility of the *acrB*-mutated BDM73185 resistant strains is unaltered, suggesting that AcrB activity is not affected at functional level, or at the level of its expression. Genetic inactivation of the *acrB* also prevents erythromycin resistance and EPI-mediated boosting of erythromycin suggesting that for erythromycin most of the boosting is mediated through AcrB inhibition. This is further supported by the lack of antibiotic boosting in the BDM73185 resistant mutants (either selected or implemented by reverse genetics) where non-AcrB pumps would, theoretically, still be inhibited. Overall however, we do not exclude that other pumps are inhibited by the pyridylpiperazines. In *E. coli* the most prominently expressed and active RND pump is AcrAB-TolC, and our data clearly demonstrated that the pyridylpiperazines act on this efflux pump.

3) Line 105, pyridomycin is introduced as a good substrate of AcrAB-TolC efflux pump. Is there evidence for this, as pyridomycin has been described as an anti-tuberculosis drug. Is pyridomycin exported exclusively by AcrAB-TolC pump? Or is there any other pump that could be involved?

Response: It is indicated that the MIC of pyridomycin is heavily improved when *tolC* or *acrB* is knocked out, a strong indication that pyridomycin is a good AcrB substrate (typical phenotype for AcrB substrates) [“antibiotic pyridomycin (1), a particularly good substrate of the AcrAB-TolC efflux pump (MIC for *E. coli* BW25113 is 12.5-25 µg/mL whereas for *E. coli* BW25113 Δ *acrB* or Δ *tolC* mutants it is 0.78 µg/mL).”]. It is not known if other pumps, expressed or not expressed, can pump pyridomycin. However, the data generated using the selected AcrB single-substitution variants of *E. coli* clearly indicate that the boosting phenotype seen by the addition of the EPI is due to the inhibition of AcrB, and this inhibition restores the bacterial susceptibility to that of the *acrB* knockout.

4) In the supplementary information method section screening library and liquid MIC determination, is there a particular reason the authors chose to dilute the *E. coli* BW25113 stock to OD600 of 0.0004 and 0.001?

Response: There is no reason for this other than historical differences in Standard Operating Procedures for screening and validation assays. This small difference in starting OD has no impact on the results of the assays, on basis of experience.

5) Figure S7 does not appear to be included properly in the document – the figure legend describes a cartoon representation of AcrB not illustrated here.

Response: We regret that this figure in the supplementary data could not be correctly visualised by the reviewer. In our file, the image appeared correctly. We have verified that it is clearly visualized also in the revised version of the manuscript.

6) The quality of Figure S5 could be improved.

Response: We interpret this comment to concern the quality/resolution of the Fig S5. Again, we regret that the reviewer was not able to correctly visualise this figure. In verification of the file, we did not observe any problem with the quality of the images in Fig S5. We have verified that the image resolution is good in the revised version of the manuscript.

7) Lines 227-230: It would be helpful to explain from the structural analysis how these residues are important in proton binding/coupling and how the inhibitor interferes with this process.

Response: In the discussion, we describe the putative interference of the inhibitor with the proton translocation in the wildtype AcrB: “From the identified inhibitor binding pocket, the piperazine moiety of the pyridylpiperazine forms a critical salt bridge with the side chain of D408. D408, together with D407, K940, and R971 are essential catalytic residues of the AcrB proton relay ((1–3) below), and the interaction of the pyridylpiperazine with D408 likely interferes with the H⁺-translocation pathway.” While it may be of interest to understanding how the substitutions inactivate the AcrB pump, this is beyond the scope of the current manuscript, where the focus is on those substitutions that do not impact AcrB function, but only affect the inhibition by the efflux pump inhibitor.

1. Tikhonova EB, Zgurskaya HI. 2004. AcrA, AcrB, and TolC of Escherichia coli form a stable intermembrane multidrug efflux complex. *J Biol Chem* 279:32116–32124.
2. Seeger MA, Von Ballmoos C, Verrey F, Pos KM. 2009. Crucial role of Asp408 in the proton translocation pathway of multidrug transporter AcrB: Evidence from site-directed mutagenesis and carbodiimide labeling. *Biochemistry* 48:5801–5812.
3. Nikaido H, Takatsuka Y. 2009. Mechanisms of RND multidrug efflux pumps. *Biochim Biophys Acta - Proteins Proteomics* 1794:769–781.

8) Does the BDM88855 compound have any NMR spectral features that might undergo chemical shift changes when bound to AcrB? If it does, this might be exploited in the future for NMR analysis of in vivo binding.

Response: Thank you for this suggestion. We could indeed have performed Saturation Transfer Difference (STD) NMR experiments with AcrB to confirm the binding of the compounds to the protein but this would have required a new collaboration. Instead, thermal shift assays have now been performed that clearly show a concentration dependent stabilisation of the AcrB/inhibitor complex and confirm a physical interaction of inhibitor with AcrB.

Minor points;

Line 48 "as a critical priority"

Response: the “a” has been added

Line 86 Image of this and how it compares to the other compounds mentioned in the above paragraph might be helpful, just to give a visual guide what EPI look like.

Response: Such an image is shown in the graphical abstract. For this reason and for the sake of limiting the number of figures (and legends) in the manuscript, we opted for omitting such a comparison in the Introduction section.

Line 116 It doesn't appear to mediate boosting in Δ acrA either - why not mention that?

Response: this has now been added:

Changes made:

“did not mediate antibiotic boosting in *E. coli* Δ acrB, Δ acrA or Δ tolC mutants (Table S3)”

Line 115 why were both BDM88855 and BDM88855.HCl tested?

Response: Compound BDM88855 was synthesized as a free base (compound 9) and as an HCl salt (compound 9'). Both compounds were tested to confirm that there was no difference in activity, which can occur due to changes in compound solubility.

Line 248 Difficult to see green rectangle with blue dots as they are so small in the figure.

Response: We regret that there seem to again have been issues with the viewing of Figure S7 (now S9) that is mentioned in line 248. As mentioned for comment 5 above, in our file, the image appeared correctly and we could not identify any green rectangles or blue dots. We have verified that it is clearly visualized also in the revised version of the manuscript.

Table 1 might be subdivided a bit more clearly e.g. different background colour for 1-7, 8, 9 and 9'-12. The naming is a bit unclear. Why are 2-7 just unnamed variants of BDM73185 but 8 is BDM88832? Was this a pre-existing name for it? Or one given later as it appeared to be a good EPI?

Response: Table 1 has been modified and sub-divided into 3 parts to show the modifications of the atom A (compounds 1 to 4), of the substituent R2 (compounds 5 to 8) and of the substituent R1 (compounds 9 to 12).

The three most interesting compounds (1, 8 and 9) which were used in several experiments have a BDM code: the hit compound BDM73185 (1) was used to select spontaneous *E. coli* resistant mutants, compound BDM88832 (8) was co-crystallized with AcrB R971A variant, compound BDM88855 (9) was co-crystallized with AcrB and tested with a panel of antibiotics.

Table S2 Confidence limits would be very helpful to determine the significance of these changes.

Response: This table presents the data of one MIC evaluation. As this work was performed with the initial hit compound, with the knowledge that the more potent compounds needed to be synthesized, it was only performed once for the whole panel of antibiotics. For the more potent BDM88855, assays were repeated with >3 replicates as indicated.

Why was BDM73185 tested against a smaller range of antibiotics compared to BDM88855.HCl?

Response: As BDM88855.HCl has much improved activity compared to the original BDM73185, we saw no benefit to performing a full screen with the less potent inhibitor.

[Please see also an attached PDF file with suggested changes to improve the presentation]

Response: We thank the reviewer for their concerted effort to improve the syntax of the manuscript. As can be seen in the revised manuscript, many of these suggestions have been implemented.

REVIEWERS' COMMENTS

Reviewer #1 (Remarks to the Author):

My concerns have been basically solved in the revised manuscript. I recommend publishing this article.

Reviewer #2 (Remarks to the Author):

The authors responded to the questions largely satisfactorily and revised their manuscript accordingly. I have no other questions.

Reviewer #3 (Remarks to the Author):

The authors have provided compelling responses to the reviewer queries and have made extensive changes to the manuscript in light of the comments.

One minor query in regard to the updated supplementary section, in the subsection 'Diffraction data collection and refinement',
"Polder electron density", is this a typo?

A point-by-point response to the reviewers' comments for
Nature Communications manuscript NCOMMS-21-26975A

REVIEWERS' COMMENTS

Reviewer #3 (Remarks to the Author):

One minor query in regard to the updated supplementary section, in the subsection 'Diffraction data collection and refinement',
"Polder electron density", is this a typo?

Response. This is not a typo. Polder electron density maps have been introduced by Liebschner et al., 2017 (PMID: 28177311), and are also known as „polder maps“ or "polder OMIT maps“. This was appropriately referenced in the manuscript.